# Safety and Efficacy of Dexmedetomidine for Bronchoscopy: A Systematic Review and Meta-Analysis

**DOI:** 10.3390/jcm12041607

**Published:** 2023-02-17

**Authors:** Qianqian Guo, Qi An, Lin Zhao, Meng Wu, Ye Wang, Zhenggang Guo

**Affiliations:** Department of Anesthesiology, Peking University Shougang Hospital, Beijing 100144, China

**Keywords:** dexmedetomidine, bronchoscopy, hypoxemia, meta-analysis

## Abstract

(1) Background: Anesthetic sedatives are widely used for bronchoscopy, and controversy surrounds the safety and efficacy of dexmedetomidine compared to other sedatives. The aim of this study is to evaluate the safety and efficacy of dexmedetomidine in bronchoscopy through a systematic review. (2) Methods: PubMed, Embase, Google Scholar, and Cochrane Library electronic databases were searched for a randomized controlled study of dexmedetomidine (Group D) or other sedative drugs (Group C) for bronchoscopy. Data extraction, quality assessment, and risk of bias analysis were performed in accordance with the preferred reporting items for systematic review and meta-analysis requirements. Meta-analysis was performed using RevMan 5.2. (3) Results: Nine studies were included, with a total of 765 cases. Compared to Group C, the incidence of hypoxemia (OR = 0.40, 95% CI (0.25, 0.64) *p* = 0.0001, I^2^ = 8%) and tachycardia (OR = 0.44, 95% CI (0.26,0.74), *p* = 0.002, I^2^ = 14%) were lower, but bradycardia (OR = 3.71, 95% CI (1.84, 7.47), *p* = 0.0002, I^2^ = 0%) was higher in Group D; no significant difference was observed in other outcome indicators. (4) Conclusions: Dexmedetomidine reduces the incidence of hypoxemia and tachycardia during bronchoscopy but is more likely to provoke bradycardia.

## 1. Introduction

Bronchoscopy is often used in the diagnosis and treatment of multiple lung diseases; flexible bronchoscopy is the most direct and effective method to check the airway [1]. An invasive procedure, bronchoscopy often causes complications, such as cough, pain, dyspnea, and other adverse reactions, and patient satisfaction with the process is poor [2,3]. Therefore, bronchoscopies are often performed in conscious sedation or even in unsedated patient [4]. Some unsedated patients can not cooperate in the examination process, which leads to the interruption of the doctor’s operation, and there is the risk of misdiagnosis and missed diagnosis. Anesthesia sedation can reduce patient discomfort, fear, eliminate unpleasant memories, create a smoother transition to operation, and improve the satisfaction of patients and bronchoscopists [4,5,6]. Conscious sedation is used even in procedures like EBUS-TBNA and it demonstrated efficacy and safety [7]. Sedatives commonly used for bronchoscopy include benzodiazepines, propofol, dexmedetomidine, and opioids [8]. Benzodiazepines enhance central inhibitory neurotransmitters, and the classical drug midazolam, which is the most widely used, has a quick onset, peak effect, and short half-life [1,9]. However, some studies show that benzodiazepines create a higher choking risk than opioids and that anesthesia consciousness requires more time to recover [10]. Opioids are also commonly used for bronchoscopy but less often for bronchoscopy alone because of the higher risk of respiratory depression [8]. Propofol is becoming increasingly common, which needs the presence of an anesthesiologist in some countries, and it is often used in combination with opioids; yet, a high-dose infusion can easily cause hypoxemia [11,12,13,14]. The use of these drugs for bronchoscopy prolongs recovery and increases the incidence of hypoxemia [15,16].

Dexmedetomidine is a drug which is actually US-FDA approved for procedural sedation, which also included bronchoscopy [17]. Dexmedetomidine acts as a novel highly selective α2-adrinephrine receptor agonists, with the effects of sedation, analgesia, memory suppression, sympathetic inhibition, and no respiratory depression, and has been shown to favor flexible bronchoscopy [18]. Compared to other sedatives, dexmedetomidine has no effect on PetCO2 but improves oxygenation [19,20,21]. Liao et al. find that compared to midazolam, the incidence of hypoxia with dexmedetomidine is significantly reduced [22]. Another study finds that compared to midazolam, dexmedetomidine improves patients’ tolerance to the test [23]. Ryu et al. find that the incidence of hypoxemia was significantly reduced, but bronchoscopist satisfaction is poor compared to fentanyl [24].

In conclusion, compared to other anesthetic drugs, whether dexmedetomidine is more suitable for flexible bronchoscopy is inconclusive. We performed a systematic review and meta-analysis on the safety and efficacy of dexmedetomidine in patients undergoing bronchoscopy.

## 2. Materials and Methods

According to the recommendations of the Cochrane Collaboration, a quantitative systematic review was conducted. The first step in this process is to define the research question by the PICO strategy (Table 1). The meta-analysis was conducted according to the systematic review and meta-analysis priority report entry (Preferred Reporting Items for Systematic Reviews and Meta-Analyses, PRISMA) [25]. Our system evaluation scheme (PROSPERO) is registered with registration number CRD 42020210380 (https://www.crd.york.ac.uk/prospero/display_record.php?ID=CRD42020210380 (accessed on 22 October 2021)).

### 2.1. Search Strategy

Employing the PubMed, Embase, Google Scholar, and Cochrane Library electronic databases, we retrieved all randomized controlled studies on dexmedetomidine for bronchoscopy from the library establishment to 31 December 2021. Relevant studies and references of the included studies were manually searched. If the full text, relevant data, and study results are incomplete, then various means of communication are used, such as e-mail addresses, to contact the authors of the studies and request the detailed raw data. The search terms were subject and associated words for dexmedetomidine, bronchoscopy, and RCTs, and the search range was obtained without language limits.

### 2.2. Inclusion/Exclusion Criteria

Studies were included in the present meta-analysis if they met the following inclusion criteria: ① patients undergoing bronchoscopy in outpatient clinics; ② a randomized controlled study comparing dexmedetomidine with other anesthetic drugs for bronchoscopy and without language limitation; ③ bronchoscopy is flexible bronchoscopy rather than rigid bronchoscopy or other types of bronchoscopy; ④ the outcomes of studies must include specific data on respiratory and cardiovascular complications and patient and bronchoscopist satisfaction. Studies with the following conditions were excluded: ① non-bronchoscopy, including bronchial ultrasound, bronchial needle aspiration, and bronchoscopic foreign body removal; ② different doses or dosing methods of dexmedetomidine; ③ review, newsletters, and abstracts; ④ studies with incomplete data; ⑤ repeated publication of the studies.

### 2.3. Literature Screen

The retrieved studies were screened by two authors (Qianqian Guo and Qi An) according to the inclusion and exclusion criteria and included in the analysis. If the two researchers disagreed about the inclusion of the study, another experienced researcher (Zhenggang Guo) was asked to evaluate it by understanding the point of debate and examining it according to the inclusion and exclusion criteria to decide whether it should be included in the analysis. Primary outcomes were hypoxemia, hypertension, hypotension, tachycardia, and bradycardia. Secondary outcomes were patient satisfaction, cough, laryngospasm, bronchospasm, and arrhythmia.

### 2.4. Data Collection

Two authors (Qianqian Guo and Qi An) independently extracted data from the final included studies, and then reviewed by the senior author (Zhenggang Guo). To reduce subjective bias, the name of the author(s), publication year, and country were temporarily withheld, and the other data were finally extracted. The content was uniformly extracted according to the form made in advance in an Excel worksheet, including general information: the name of the author(s), date of publication, and the basic characteristics of the included studies: number of patients, age, weight, ASA grade, operation type, intervention characteristics and inclusion indicators.

### 2.5. Bias Assessment

The methodological quality of the included studies was independently evaluated by two authors (Qianqian Guo and Qi An) according to the RCT evaluation criteria recommended by the Cochrane Intervention System Evaluation Manual (Cochrane Handbook for Systematic Reviews of Interventions) [26]. If the two researchers disagreed about the quality of the study, another experienced researcher (Zhenggang Guo) was asked to evaluate. Specific content included random sequence generation (selective bias), allocation concealment (selective bias), blinding of participants and personnel (performance bias), blind outcome assessment (detection bias), incomplete outcome data (attrition bias), selective reporting (reporting bias), and other biases. The attrition bias included pre-intervention baseline measurements and post-intervention effect parameter values, loss to follow-up/dropout (whether loss to follow-up rate is 10%), data from exclusion analysis, whether to explain the reasons for the loss to follow-up, and whether the intentionality analysis of the patients lost to follow-up was conducted. For each included study, low (low risk), high (high risk), or unclear risk (unclear risk) (lack of relevant information or uncertain bias) were judged according to the above seven items.

### 2.6. Meta-Analysis

Meta-analysis was performed using RevMan 5.2. For binary variables, odds ratio (OR), relative risk (RR) and 95% confidence interval (CI) was used. Forest plots, OR or RR and 95% CI were represented as the results, and *p* < 0.05 was considered statistically significant. The χ^2^ and I^2^ tests were performed for clinical heterogeneity in the included studies, and *p* < 0.10 and I^2^ > 50% showed that χ^2^ had statistical differences. If the outcomes of the studies had low heterogeneity (*p* > 0.10, I^2^ < 50%), a fixed-effects model was selected and OR and 95% CI were represented as the results. Otherwise, the random effects model was selected for meta-analysis and RR and 95% CI were represented as the results. Subgroup analysis and sensitivity analysis excluding literature one by one were used to explore the causes of high heterogeneity.

## 3. Results

### 3.1. Literature Search and Basic Characteristics of the Literature

The electronic databases were explored according to the search strategy, resulting in 2040 studies, with 93 duplicate studies removed. A total of 1924 studies were excluded after reading literature titles and abstracts. Fourteen studies did not meet the inclusion criteria. Finally, nine studies were included with 765 patients (see Figure 1).

All studies were randomized controlled studies of flexible bronchoscopy. Dexmedetomidine loading dose was 1 µg/kg in five studies [22,23,27,28,29], 0.5 µg/kg in three studies [24,30,31], and 2 µg/kg in one study [32]. The included studies of dexmedetomidine were administered intravenously, with the exception of one study [32], which was administered by aerosol. The basic characteristics of the included studies are listed in Table 2.

### 3.2. Quality Evaluation

The risk of bias assessment for included studies is shown in Figure 2. All the studies demonstrated a low risk in terms of random sequence generation, incomplete outcome data, and selective reporting. For a small number of studies with assigned sequence hidden [22,29,32], subjects and researchers blinded [22], and blinded outcome measures, the researchers were unable to assess the risk [27,29,32]. Only one study because of allocation concealment, subjects, and operators’ implementation blindness was unclear, and the outcome measures were assessed by the operators, so the results are high risk [22].

### 3.3. Results of the Meta-Analysis

#### 3.3.1. Hypoxemia

Nine studies have explored the relationship between the use of anesthetic drugs and hypoxemia during flexible bronchoscopy, including a total of 765 patients [22,23,24,27,28,29,30,31,32]. Hypoxemia is defined as displaying SpO2 < 90% for more than 30 s, and it can be controlled by increasing the oxygen flow rate to 6 L/min or by performing mandibular support if necessary. Compared with the control group (Group C), the dexmedetomidine group (Group D) was less prone to hypoxemia (OR = 0.40, 95% CI (0.25, 0.64), *p* = 0.0001, I^2^ = 8%), and the difference was statistically significant, meanwhile the heterogeneity between groups was low (Figure 3). Due to control drugs of the including studies involved propofol, midazolam, lidocaine, and opioids, meta-analysis was performed for subgroup analysis. Dexmedetomidine vs. midazolam (*p* = 0.26) and dexmedetomidine vs. lidocaine (*p* = 0.49) were not statistically significant. Dexmedetomidine vs. propofol (*p* = 0.02) and dexmedetomidine vs. opioids (*p* = 0.03) were statistically significant (Figure 4).

#### 3.3.2. Hypertension

Five studies explored hypertension, including a total of 477 patients [22,24,27,29,32]. Hypertension is defined as a systolic blood pressure (SBP) > 180 mmHg, diastolic blood pressure (DBP) > 100 mmHg, or blood pressure 20% higher than the baseline arterial pressure. Compared with Group C, Group D experienced less hypertension, the difference was not statistically significant. There was low heterogeneity among the included studies (OR = 0.53, 95% CI (0.28,1.02), *p* = 0.06, I^2^ = 0%) (Figure 5).

#### 3.3.3. Hypotension

Five studies discussed hypotension, including a total of 440 patients [23,24,27,29,32]. Hypotension is defined as SBP < 90 mmHg or mean arterial pressure < 60 mmHg or 20% below the baseline arterial pressure, which can be treated with ephedrine 5–10 mg injection [24,29]. The risk of hypotension in Group D and Group C was similar (OR = 1.19, 95% CI (0.61, 2.32), *p* = 0.62, I^2^ = 0%). The heterogeneity between the included studies was low, and there was no statistical difference (Figure 6).

#### 3.3.4. Tachycardia

Five studies discussed tachycardia, including a total of 477 patients [22,24,27,29,32]. Tachycardia is defined as a heart rate (HR) > 100 beats/minute and/or a change of >20% compared with the baseline value. It can be treated by injecting 2 mL 2% lidocaine into the trachea or increasing the dosage of the anesthetic [24,29]. Compared with Group C, Group D was less likely to experience tachycardia during bronchoscopy (OR = 0.44, 95% CI (0.26, 0.74), *p* = 0.002, I^2^ = 14%), which was statistically significant, and the heterogeneity between the included studies was low (Figure 7).

#### 3.3.5. Bradycardia

Six studies discussed bradycardia, including a total of 553 patients [22,23,24,27,29,32]. The definition of bradycardia is the presence of HR < 50–60 beats/min or 20% lower than the baseline heart rate, which is treated with 0.3–0.5 mg or 0.01 mg/kg atropine [24,28,29]. Compared with Group C, Group D was more likely to have bradycardia during bronchoscopy (OR = 3.71, 95% CI (1.84, 7.47), *p* = 0.0002, I^2^ = 0%), which was statistically significant, and the heterogeneity between included studies was low (Figure 8).

#### 3.3.6. Patient Satisfaction Level

Five studies discussed patient satisfaction, including a total of 350 patients [23,24,29,30,32]. There was no statistically significant difference between the satisfaction of patients in Group D and Group C during bronchoscopy (RR = 1.29, 95% CI (0.90, 1.68), *p* = 0.2, I^2^ = 84%), and the heterogeneity between the included studies was high (Figure 9).

#### 3.3.7. Arrhythmia

Two studies discussed arrhythmia [24,27]. Compared with Group C, Group D had fewer arrhythmias during bronchoscopy (OR = 0.61, 95% CI (0.15, 2.48), *p* = 0.49, I^2^ = 0%). The difference was not statistically significant, and the heterogeneity between the included studies was low (Figure 10).

#### 3.3.8. Cough

Two studies discussed cough [27,31]. Group D was more likely to cough during bronchoscopy than Group C (RR = 1.65, 95% CI (0.63, 4.29), *p* = 0.31, I^2^ = 83%). However, the difference was not statistically significant, and the heterogeneity was high (Figure 11).

#### 3.3.9. Laryngospasm

Only one study explored laryngospasm [32], and one patient in Group C demonstrated laryngospasm during the examination.

#### 3.3.10. Bronchospasm

None of the patients in the included studies developed bronchospasm.

## 4. Discussion

This is the first systematic review and meta-analysis comparing the safety and effectiveness of dexmedetomidine in bronchoscopy. This systematic review and meta-analysis included a total of nine studies, including 765 patients. The meta-analysis founds that dexmedetomidine significantly reduces the incidence of hypoxemia and tachycardia during bronchoscopy compared with control group. The results of the nine included studies were consistent, and the heterogeneity was low. However, dexmedetomidine significantly increased the incidence of bradycardia compared with the control group. There were no significant differences between the two groups in hypertension, hypotension, or patient satisfaction.

Bronchoscopy is an invasive procedure often used to diagnose and treat respiratory diseases. Currently, guidelines recommend the use of sedatives during the examination of patients without contraindications, which can enhance patient tolerance and reduce the occurrence of complications [4]. Common anesthetic tranquilizers include benzodiazepines, opioids, propofol, ketamine, anticholinergics, and other drugs used alone or in combination with others [8]. Many anesthetic schemes are available, each with advantages and disadvantages in terms of the depth of sedation, blood oxygen saturation, and hemodynamics.

Benzodiazepines such as midazolam are commonly used. Compared with propofol, they have the characteristics of cis amnesia but have a slow onset, long recovery time, and only a sedative effect, lacking analgesic effect, and coughing incidence is high. They are often used in combination with opioids. Propofol has a quick onset and drug metabolism and is often used in combination with opioids. It is worth noting that the use of propofol needs to be continuously monitored by anesthesiologists. Because the therapeutic window between moderate anesthesia and general anesthesia is relatively narrow, and there is no available antagonist, when given in large doses, propofol has a serious inhibitory effect on respiration and circulation, increasing the incidence of hypoxemia and hypotension [8]. Dexmedetomidine is a relatively new sedative used in bronchoscopy [33]. As a highly selective α-2 adrenaline receptor agonist and due to its unique pharmacological properties, dexmedetomidine can provide sufficient sedation and analgesia in conscious intubation of difficult airways and fibro-bronchoscopy examination of ICU patients without inhibiting the respiratory function of patients [34,35,36].

The results of this meta-analysis showed that dexmedetomidine had a lower incidence of hypoxemia compared with the control group (*p* = 0.0001). By subgroup analysis of control drugs, propofol and opioids increased the incidence of hypoxemia during flexible bronchoscopy, but midazolam and lidocaine had no significant difference in the incidence of hypoxemia compared with dexmedetomidine. This means that dexmedetomidine has less effect on breathing during flexible bronchoscopy, and more attention should be paid to patients’ oxygen intake when using propofol and opioids. Of note, dexmedetomidine was more likely to cause bradycardia than control drugs, and the difference was statistically significant (*p* = 0.0002), which may be related to its sympathetic inhibition. Although it can be corrected by giving atropine, it should be selected using caution for elderly patients, or patients with a high risk of cardiovascular events such as conduction block and arrhythmia.

Among the included studies, five studies recorded the number of patients with hypotension [22,23,24,28,29]. There was no significant difference in the incidence of hypotension between dexmedetomidine and control drugs (*p* = 0.62), and the heterogeneity was low. Similarly, there were five studies that counted the number of patients with hypertension [22,24,27,29,32]. There was no significant difference in the incidence of hypotension between dexmedetomidine and control drugs (*p* = 0.06), and the heterogeneity was low. The results showed that there was no statistical difference between dexmedetomidine and control drug on hemodynamics in patients undergoing flexible bronchoscopy.

Because the evaluation criteria for patient satisfaction in the included studies were inconsistent, we formulated unified rules for the different evaluation criteria. Patients who clearly expressed satisfaction in the study and those who expressed willingness to accept the bronchial examination again were deemed to be satisfied with this examination; otherwise, they were deemed to be dissatisfied. The results show that there was no difference in satisfaction between Group D and Group C (*p* = 0.2), but the heterogeneity of the literature was high, which may be inconsistent with the evaluation criteria of satisfaction. The subjectivity of the patients’ satisfaction expression was strong. So more clinical evidence is required to prove it.

In two studies [24,27], there were eight patients with arrhythmia, one of whom was in Group C, but the type of arrhythmia was not described, which did not cause serious consequences [27]. In another study [24], 3 patients in Group D and four patients in the remifentanil group had atrial or ventricular premature contractions that subsided spontaneously without hemodynamic instability. Because the original study did not analyze the cause of arrhythmia, the sample size of the study was small, the number of positive cases was also small, and the difference was not statistically significant, making it impossible to determine the specific cause of arrhythmia, which may be related to the operator, individual specificity, insufficient depth of sedation and analgesia, cough or secretion, or other stimuli, and further research is needed.

In one study, there was one case of laryngospasm in Group C [32]. Two studies observed cough, one of which showed that patients in Group D were more likely to cough, and the difference was statistically significant. However, due to fewer studies, small sample size, and insufficient test efficiency, it is impossible to draw a conclusion, and more clinical evidence is needed to further prove it [27,31].

There are several limitations in this meta-analysis. First, a total of nine studies were included in this meta-analysis. Although all studies included hypoxemia, only a few studies included other outcome indicators, which may affect the reliability of the results. Secondly, due to limited studies on other indicators, subgroup analysis of these outcome indicators for different drugs could not be conducted. Thirdly, there is a study on children, who are different from adults in physiology and sensitivity to drugs, so meta-analysis may be biased. Finally, it is hoped that more randomized controlled studies with large samples and multi-centers can verify our conclusions in the future.

## 5. Conclusions

During bronchoscopy, the use of dexmedetomidine can significantly reduce the incidence of hypoxemia in patients; however, it is necessary to be alert to the occurrence of bradycardia.

## Figures and Tables

**Figure 1 jcm-12-01607-f001:**
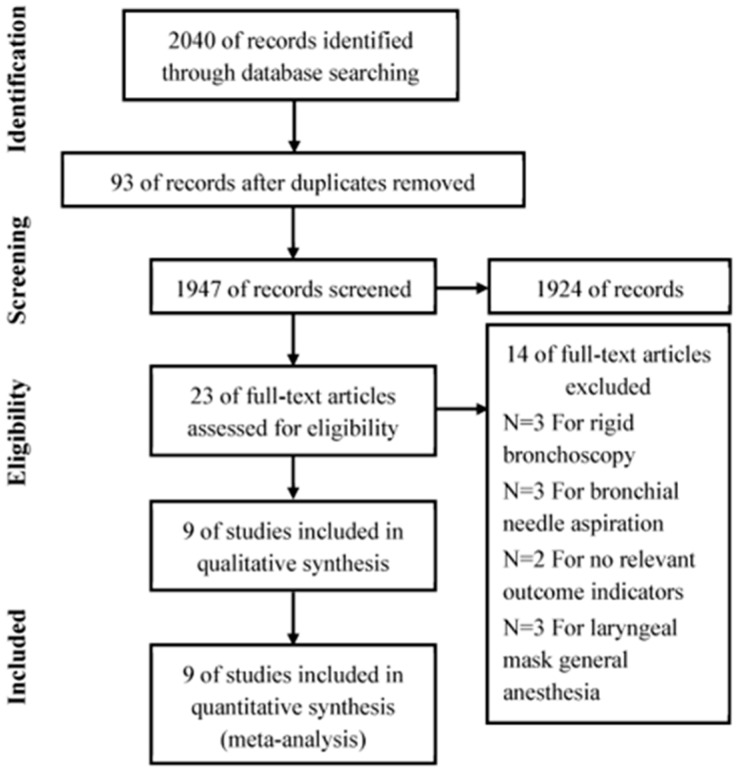
Flow chart of article selection for the meta-analysis.

**Figure 2 jcm-12-01607-f002:**
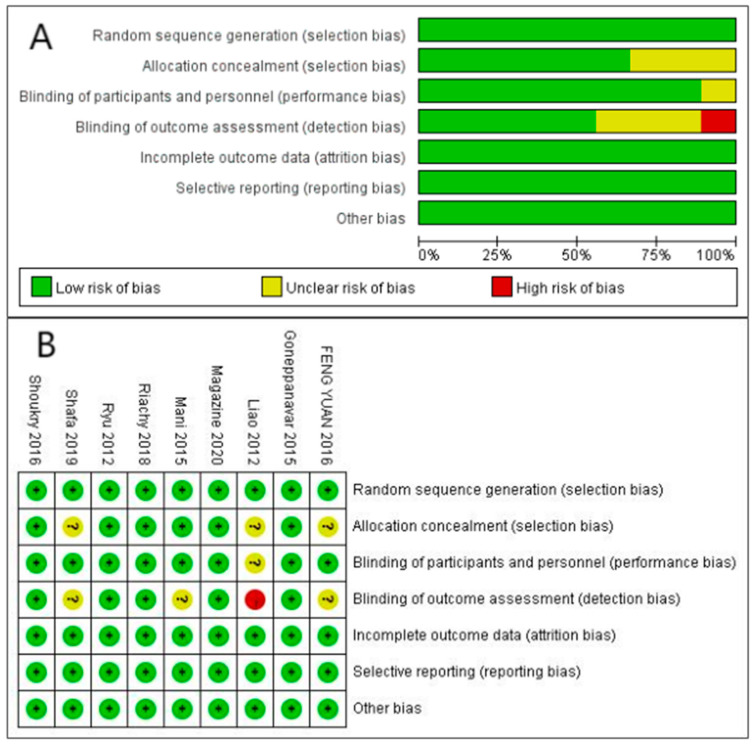
Risk of bias assessment. (**A**): Risk of bias summary. (**B**): Risk of bias graph. Green means low risk. Red means high risk. Yellow means unclear risk [22,23,24,27,28,29,30,31,32].

**Figure 3 jcm-12-01607-f003:**
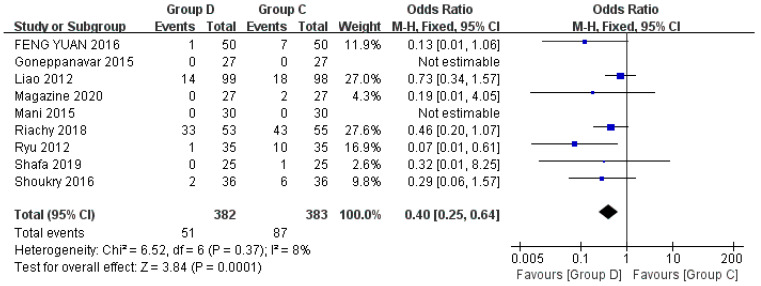
Forest plots comparing the effects of dexmedetomidine and control group on hypoxemia [22,23,24,27,28,29,30,31,32].

**Figure 4 jcm-12-01607-f004:**
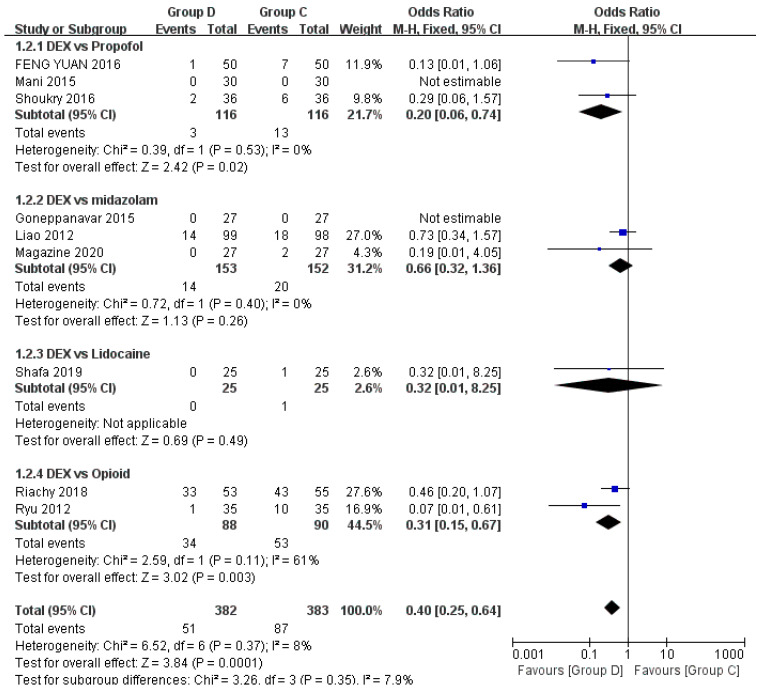
Forest plots of subgroup analysis comparing the effects of dexmedetomidine and control group on hypoxemia [22,23,24,27,28,29,30,31,32].

**Figure 5 jcm-12-01607-f005:**
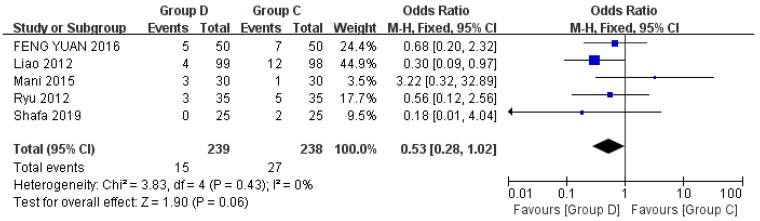
Forest plots comparing the effects of dexmedetomidine and control group on hypertension [22,24,27,29,32].

**Figure 6 jcm-12-01607-f006:**
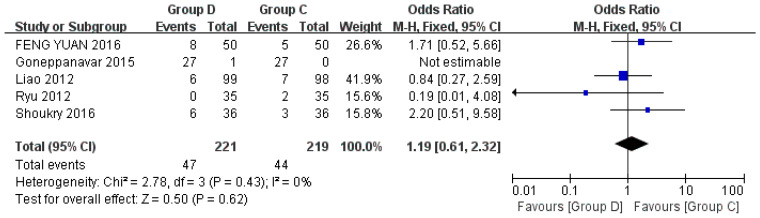
Forest plots comparing the effects of dexmedetomidine and control group on hypotension [22,23,24,28,29].

**Figure 7 jcm-12-01607-f007:**
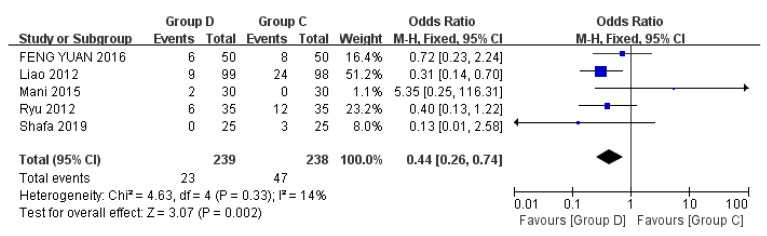
Forest plots comparing the effects of dexmedetomidine and control group on tachycardia [22,24,27,29,32].

**Figure 8 jcm-12-01607-f008:**
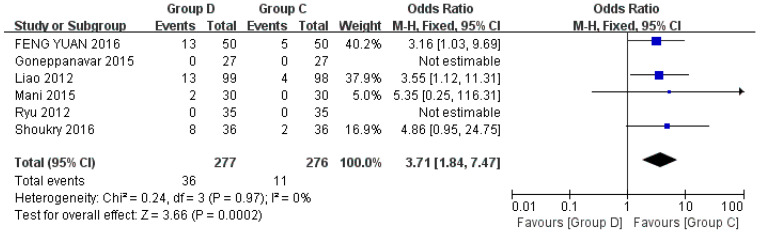
Forest plots comparing the effects of dexmedetomidine and control group on bradycardia [22,23,24,27,28,29].

**Figure 9 jcm-12-01607-f009:**
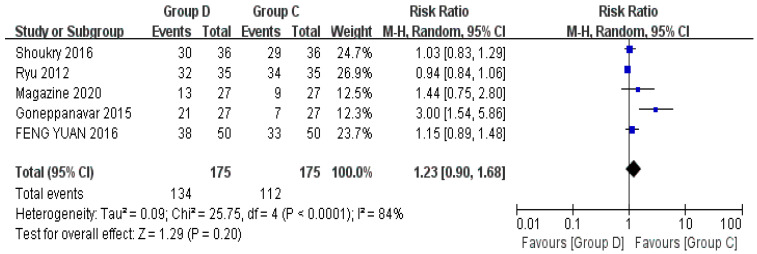
Forest plots comparing the effects of dexmedetomidine and control group on patient satisfaction level [23,24,28,29,30].

**Figure 10 jcm-12-01607-f010:**
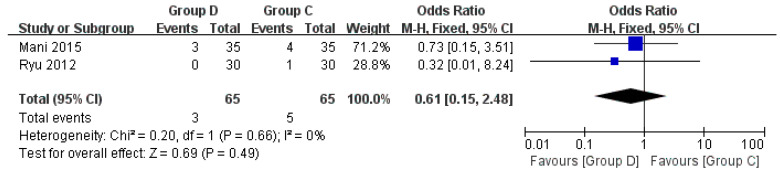
Forest plots comparing the effects of dexmedetomidine and control group on arrhythmia [24,27].

**Figure 11 jcm-12-01607-f011:**
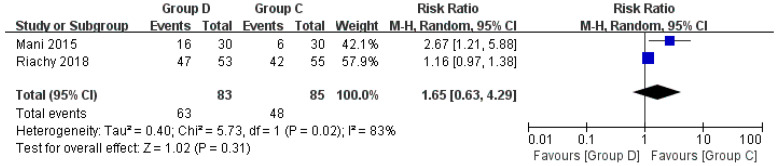
Forest plots comparing the effects of dexmedetomidine and control group on cough [27,31].

**Table 1 jcm-12-01607-t001:** PICO format.

Category	Description
Population	patients undergoing bronchoscopy in outpatient clinics
Intervention	Anesthetic drugs are used alone or in combination to maintain continuous sedation during bronchoscopy
Comparison	Dexmedetomidine vs. other anesthetic drugs
Outcomes	Primary outcomes were hypoxemia, hypertension, hypotension, tachycardia, and bradycardia. Secondary outcomes were patient satisfaction, cough, laryngospasm, bronchospasm, and arrhythmia.

**Table 2 jcm-12-01607-t002:** Basic characteristics of the included studies.

Study, Year	Number of Patients		ASA (I/II/III)	Age (years)	Height (cm)	Weight (kg)	BMI (kg/m^2^)	Intervention		Administration Route	Sedation Score
	G D	G C						G D	G C		
FENG YUAN, 2016 [29]	50	50	42/29/29	60.11 ± 7.27	N/A	N/A	21.95 ± 3.11	1 µg/kg Dexmedetomidine + 1 µg/kg fentanyl	4 µg/mL propofol + 1 µg/kg fentanyl	Intravenous infusion	MOAA/S
Shoukry, 2016 [28]	36	36	N/A	49.1 ± 15.1	N/A	79.1 ± 14.1	N/A	1 µg/kg Dexmedetomidine + 1 µg/kg fentanyl	0.5–1 mg/kg propofol + 1 µg/kg fentanyl	Intravenous infusion	Ramsay Sedation Score
Shafa, 2019 [32]	25	25	N/A	2.34 ± 1.77	N/A	N/A	N/A	2 µg/kg Dexmedetomidine	1% Lidocaine (4 mg/kg Nebulized)	aerosol	Ramsay Sedation Score
Ryu, 2012 [24]	35	35	47/23	52.9	163	61.35	N/A	0.5 mg/kg propofol + 0.4–2 µg/kg Dexmedetomidine	0.5 mg/kg propofol + 1–5 µg/kg remifentanil	Intravenous infusion	MOAA/S
Riachy, 2018 [31]	53	55	N/A	18–70	N/A	N/A	N/A	0.5 µg/kg Dexmedetomidine	10 µg/kg alfentanil	Intravenous infusion	NICS
Mani, 2015 [27]	30	30	39/21	44.75 ± 14.8	164.9	61.3 ± 6.15	N/A	1 μg/kg Dexmedetomidine	1 mg/kg propofol	Intravenous infusion	MOAA/S
Maga- zine, 2020 [30]	27	27	N/A	46.04 ± 13.83	159.92 ± 9.13	51.8 ± 9.27	20.30 ± 3.39	0.5 μg/kg Dexmedetomidine	0.035 mg/kg midazolam	Intravenous infusion	Ramsay Sedation Score
Liao, 2012 [22]	99	98	N/A	59.3 ± 8.75	161.75 ± 7.80	57.60 ± 9.84	N/A	1 µg/kg Dexmedetomidine	2 mg midazolam	Intravenous infusion	Ramsay Sedation Score
Gonepp-anavar, 2015 [23]	27	27	N/A	51.11 ± 14.45	162.45 ± 5.41	54.67 ± 10.47	20.60 ± 3.43	1 μg/kg Dexmedetomidine	0.02 mg/kg midazolam	Intravenous infusion	Ramsay Sedation Score

N/A, not available; GD: Group D, the dexmedetomidine group; GC: Group C, the control group; ASA: American Society of Anesthesiologists; BMI: Body mass index.

## Data Availability

All authors agree to data sharing, and anyone who would like more data should contact the author at shouganggqq@163.com.

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
