# Peer review of "Safety and Efficacy of Dexmedetomidine for Bronchoscopy: A Systematic Review and Meta-Analysis"

_jcm, 2023, doi:10.3390/jcm12041607_

Round 1

Reviewer 1 Report

First of all, thank you to the editor for give me the opportunity to revise this interesting article.

The paper is a systematic review to evaluate the safety and efficacy of dexmedetomidine in bronchoscopy. The topic is interesting and important.
Overall, the article is well structured.
I think that the manuscript deserves to be published, after some minor changes.

At line 34 the authors affirm that "bronchoscopy is usually performed under deep sedation". I think that this sentence is not correct. Bronchoscopies are often performed in conscious sedation or even in unsedated patient (Du Rand, I.A.; Blaikley, J.; et al. British Thoracic Society Guideline for Diagnostic Flexible Bronchoscopy in Adults. Thorax 2013). Conscious sedation is used even in procedures like EBUS-TBNA and it demonstrated efficacy and safety (Piro R, Casalini E, et al. Efficacy and safety of EBUS-TBNA under conscious sedation with meperidine and midazolam. Thorac Cancer. 2022 Feb;13(4):533-538). I think that this paragraph needs to be changed and completed.

At line 42 the authors speak about propofol as a drug increasingly common. I think it is true in some centers but not everywhere. It could be usuful to remember also in this paragraph that in some countries propofol needs the presence of an anesthesiologist.

At line 180, 189-190, 198-199 the treatements of complications are reported: a reference is needed.

The sentence at line 263-264 is repeated at line 264-266.

The sentences at line 263-275 are repeated at line 276-288.

Reviewer 2 Report

Dear authors,

Thank you for submitting your work. Kindly revise your manuscript based on following comments or provide explanation to justify if no edits possible.

1. What needs to be mentioned here is that Dex is a drug which is actually US-FDA approved for procedural sedation, which also included bronchoscopy. Also, provide a reference for the same.

2. Google Scholar is not an ideal database for searching studies for SRMA. However, I will not suggest any change now.

3. The methods section should define the research question based on PICO format: Population, Intervention, Comparison and Outcomes.

4. The authors can add sub-headings to the methods section to make easy reading like:

search strategy, inclusion/exclusion criteria, PICO, Bias assessment, meta-analysis.

5. Line 115: authors mention the use of relative risk for binary variables. It should rather be risk ratio. However, the forest plot depicts the use of Odds ratio. This is a major contradiction. This needs to be rectified.

6. The results does not have a quality assessment of evidence (GRADE tool, Jadad score). The authors are requested to provide an explanation for the same.
